# Post-Mastectomy Breast Reconstruction Disparities: A Systematic Review of Sociodemographic and Economic Barriers

**DOI:** 10.3390/medicina60071169

**Published:** 2024-07-19

**Authors:** Kella L. Vangsness, Jonathan Juste, Andre-Philippe Sam, Naikhoba Munabi, Michael Chu, Mouchammed Agko, Jeff Chang, Antoine L. Carre

**Affiliations:** City of Hope, 1500 E Duarte Rd, Duarte, CA 91010, USA; jonathan.juste@student.cusm.edu (J.J.); andre-philippe.sam@medsch.ucr.edu (A.-P.S.); naikhoba.munabi@med.usc.edu (N.M.); dr.michael.chu@gmail.com (M.C.); magko@coh.org (M.A.); jechang@coh.org (J.C.); acarre@coh.org (A.L.C.)

**Keywords:** breast reconstruction, surgery, postmastectomy breast reconstruction, disparities

## Abstract

*Background*: Breast reconstruction (BR) following mastectomy is a well-established beneficial medical intervention for patient physical and psychological well-being. Previous studies have emphasized BR as the gold standard of care for breast cancer patients requiring surgery. Multiple policies have improved BR access, but there remain social, economic, and geographical barriers to receiving reconstruction. Threats to equitable healthcare for all breast cancer patients in America persist despite growing awareness and efforts to negate these disparities. While race/ethnicity has been correlated with differences in BR rates and outcomes, ongoing research outlines a multitude of issues underlying this variance. Understanding the current and continuous barriers will help to address and overcome gaps in access. *Methods*: A systematic review assessing three reference databases (PubMed, Web of Science, and Ovid Medline) was carried out in accordance with PRISMA 2020 guidelines. A keyword search was conducted on 3 February 2024, specifying results between 2004 and 2024. Studies were included based on content, peer-reviewed status, and publication type. Two independent reviewers screened results based on title/abstract appropriateness and relevance. Data were extracted, cached in an online reference collection, and input into a cloud-based database for analysis. *Results*: In total, 1756 references were populated from all databases (PubMed = 829, Ovid Medline = 594, and Web of Science = 333), and 461 duplicate records were removed, along with 1147 results deemed ineligible by study criteria. Then, 45 international or non-English results were excluded. The screening sample consisted of 103 publications. After screening, the systematic review produced 70 studies with satisfactory relevance to our study focus. *Conclusions*: Federal mandates have improved access to women undergoing postmastectomy BR, particularly for younger, White, privately insured, urban-located patients. Recently published studies had a stronger focus on disparities, particularly among races, and show continued disadvantages for minorities, lower-income, rural-community, and public insurance payers. The research remains limited beyond commonly reported metrics of disparity and lacks examination of additional contributing factors. Future investigations should elucidate the effect of these factors and propose measures to eliminate barriers to access to BR for all patients.

## 1. Introduction

Breast reconstruction (BR) following mastectomy has been well established as a beneficial medical intervention for patients’ physical and psychological well-being for more than two decades. Numerous studies have emphasized BR as the gold standard of care treatment for breast cancer patients requiring surgical management [1,2,3]. There have been multiple policies enacted to improve access, yet there remain social, economic, and geographical barriers to receiving reconstruction. Understanding the current and continuous barriers will help to address and overcome gaps in access.

Legislative measures enacted to address disparities in reconstruction include the Women’s Health and Affordable Care Act of 1998; the Breast Cancer Protection Act of 2001, which penalizes insurance companies for non-coverage of breast cancer; and the 2011 Affordable Care Act’s expansion of Medicaid access [4,5,6]. Previous studies have demonstrated that these laws have been beneficial in increasing BR rates amongst all racial/ethnic groups over time [7]. The works of [8,9] even provided specific examples of single-institution safety-net hospitals where largely underserved, minority, and Medicaid-insured patients received BR at rates at or above the national average [8,9]. These findings provide hope in efforts to rectify disparities in breast cancer treatment disparities; however, progress has not been consistent or universal, and many distinct subpopulations continue to face unequal access to BR. Previously identified disparities have been found amongst specific ethnic groups, including African Americans, Native Americans, Hispanics, and Asians [10,11,12]. Additional groups receiving less reconstruction have been identified as those with low socioeconomic status, with non-private insurance, in rural or isolated communities, and of older age.

Interestingly, a systematic review of the study sample elucidated several additional intrinsic and extrinsic factors beyond race, insurance status, geographic location, and age that impact not only equitable access to healthcare but also the quality of breast cancer diagnosis, management, postoperative outcomes, and overall mortality nationwide. These multifactorial impediments include inconsistent preoperative communication between plastic surgeons and patients, inherent cultural biases, mistrust of the healthcare system, variability in baseline comorbid status, occupational constraints, paucity of skilled surgical specialists in proximity to patients, and differences in hospital status [13,14,15,16,17,18,19,20].

Threats to equitable and concordant quality of healthcare among all breast cancer patients in America ostensibly persist today, despite growing public health awareness within the plastic and reconstructive surgery community and deliberate efforts to negate disparities in breast cancer treatment. While race/ethnicity have been extensively characterized as significantly correlated with differences in BR rates and outcomes, ongoing research has highlighted that there is a complex multitude of issues underlying the variation in rates and outcomes of BR following mastectomy. The results of numerous preceding studies focused on controlling for socioeconomic variables and patient demographic factors helped inform our review and support a hypothesis of an evolving, abstruse interaction between many variables on patient BR treatment rates and outcomes [15,16,21,22,23,24,25,26]. This systematic review aims to comprehensively outline and specify sociodemographic, economic, and clinical factors contributing to the distribution of breast reconstruction for cancer patients.

## 2. Methods

### 2.1. Research Design

A systematic review assessing three leading biomedical and interdisciplinary reference databases, namely PubMed, Web of Science, and Ovid Medline databases, was registered in accordance with PRISMA (Preferred Reporting Items for Systematic Reviews and Meta-Analyses) 2020 guidelines on 3 February 2024 (Figure 1) under registration identification number CRD42024529746. The protocol is available through the National Institute for Health and Care Research. An amendment was made to refine the search query to include “lumpectomy”. A predefined search query was constructed from multiple keywords and Boolean operators: “rate of breast reconstruction after mastectomy OR rate of breast reconstruction after lumpectomy OR effect of Women’s Health and Cancer Rights Act on breast reconstruction OR breast reconstruction after Affordable Care Act OR rate of breast reconstruction on African Americans OR rate of breast reconstruction on Asians OR disparity in breast reconstruction”; for succinct interpretation, search parameters limited article results to those published in English by domestic institutions between the years 2004 and 2024. This study period was selected to span modestly after enactment of the 1998 WHCRA, by which time the public health impact of such legislation would presumably be comfortably measured, onwards for the twenty years (terminating with publications available at the time of this systematic review search query). It was hypothesized that this timeframe would allow for optimal observation of possible BR procedure frequency trends in response to the influence of previously described federal legislation and healthcare mandates in America. This systematic review search was dedicated to the assessment of the burgeoning record of inquiry into national healthcare disparity, specifically concerning oncologic breast surgery.

### 2.2. Search Term Definitions

Breast reconstruction encompasses the creation of a new breast after full or partial removal of the breast. This can be done at the time of mastectomy, termed “immediate breast reconstruction”, while “delayed breast reconstruction” is breast reconstruction that takes place after the mastectomy surgery. Breast reconstruction can be done “autologously” which uses the patient’s tissue to rebuild the breast, or “implant” based, which uses a breast implant. Lumpectomy is a well-accepted technical term used to describe breast-conserving therapy.

### 2.3. Selection Criteria

In addition to preliminary filters applied to the primary database search, studies were included based on peer-reviewed status and publication type. Eligible sample studies included prospective, retrospective, observational, case–control, randomized controlled trials, and qualitative analyses. Initial exclusion criteria included international articles, foreign language publications, duplicate results, editorial papers, conference abstracts, book chapters, and non-peer reviewed status. Building upon [6] established classification of American healthcare disparity across the population as a compound amalgam framework (ordinally developed by the National Institute on Minority Health and Health Disparities, NIMHD), we prioritized inclusion of studies that provide a discerning, holistic approach to evaluating/rectifying BR disparities. Evidence-based medicine may be maximally utilized for the benefit of disadvantaged populations when it is accompanied by an appreciation and awareness of the deep-seated interactions between patient demographics and various health determinants [6,27].
Figure 1PRISMA Flow Chart [28]. Comprehensive table outlined in Appendix A.
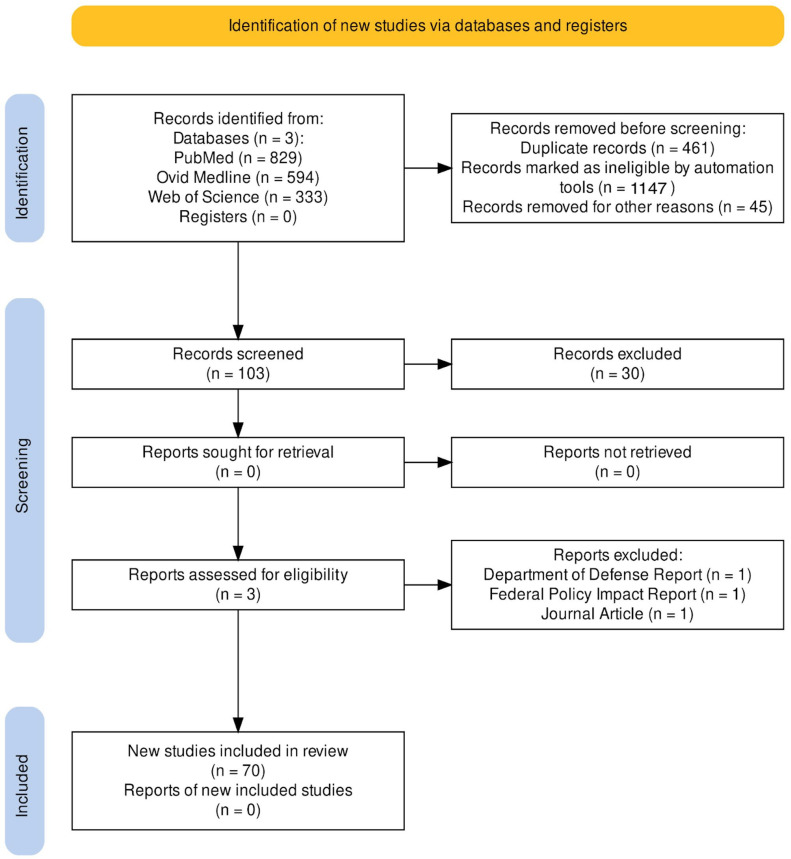


### 2.4. Sample Screening

Studies were subjected to a cursory screening by two independent reviewers based on title/abstract appropriateness and relevance to the research topic of interest. Following this refinement process, both independent reviewers individually appraised the effectiveness of each sample study via full text and reference review. Bias risk and strength of study design were specifically evaluated for each study with the assistance of Elsevier’s Assessment of Evidence. Review conflicts were resolved using a third reviewer. This method of ensuring accurate methodological data collection was to avoid bias.

### 2.5. Data Collection

Data were extracted from individual studies, cached within an online group, i.e., Zotero reference collection, and was subsequently input sequentially into an online cloud-based database for collaboration. Specific variables collected for each sample study included the following: authors, research title, PMID identifier, year of publication, publication source, study design, study setting, primary and secondary outcomes, study period, total number of study participants, demographic information collected, study inclusion and exclusion criteria, measurement of outcomes, results, *p*-value significance level and confidence intervals (CI), and possible adverse or unintended consequences of the intervention/exposure, and finally, a quality assessment grade (level of evidence) was assigned. Studies were assessed by their capacity to demonstrate significant variation in breast reconstruction rates either amongst groups, throughout time, and across geographical regions or whether breast cancer diagnosis, management, and post-operative outcomes showed significant disproportionality. Several studies also were noted to propose additional potential aggravating and alleviating factors that influenced healthcare access and outcomes, highlighting many potential effects of determinants of health.

## 3. Results

In total, 1756 references were populated from the three databases employed (PubMed = 829, Ovid Medline = 594, and Web of Science = 333). Then, 461 duplicate records were removed prior to screening, along with 1147 search results deemed ineligible for study criteria based on automated selection methods. Next, 45 results were excluded due to international origin or non-English language publication. After the initial inclusion criteria were applied to the results, the study screening sample consisted of 103 unique publications. No report-based publications were intentionally screened for inclusion in the systematic review. The two independent screeners examined full texts and references of the 103 study records and excluded 30 individual studies that did not meet study inclusion standards, with reasons including irrelevant scope of focus, editorial publication, systematic review papers that might have introduced undue sample bias, uncertain peer-review status, and report-style publication. Three reports were screened and subsequently excluded, with results presenting information from Department of Defense healthcare data, federal policy impact measures, and a research study-based journal publication. After all ineligible formats were screened and excluded, the systematic review produced 70 studies with satisfactory relevance to our study focus and collectively provided a dynamic set of insights into the discussion of BR disparity across the study period (Appendix A). The disparity type discussed within the literature was found to heavily favor sociodemographics, insurance type, and geography categories (Table 1). Data were grouped and are discussed based on disparity type to isolate contributing variables.

### 3.1. Rates of Reconstruction

Multiple national databases have shown the rate of reconstruction has increased over time across almost all variables from 2004 to 2016, from 36.3% to 63.8% [29,30,31]. Immediate breast reconstruction rates increased from 26% in 2005 to 40% in 2011 [14,32]. Compared to 1998, the odds of having an immediate postmastectomy breast reconstruction in 2006 were 2.13 times greater [33]. In a study of the National Cancer Database (NCDB) in 1998–2007, the use of immediate and early delayed postmastectomy breast reconstruction among patients nearly doubled from 13.3% to 25.6% [10]. In a large state study of New York [34] and the Nationwide Inpatient Sample (NIS) [35], there was an immediate breast reconstruction (IBR) increase from 27% to 59% between 2005 and 2015.

### 3.2. Racial Disparities

Increased rates of reconstruction were seen across all races; however, minority races overall received less reconstruction compared to White patients [2,10,34,35,36,37,38,39,40]. Reconstruction was higher among Whites/Caucasian compared to African American [11,17,21,22,39,41,42,43,44,45,46,47] patients and persisted when controlled for sociodemographic geographical variables in large sample national databases. From 2005 to 2014, there was an increase in breast reconstruction among all racial groups from 33.2% to 60% [48]. In 2014, the difference between White and Black was 8.17%, White and Asian 14.77%, and White and Native American 19.03%. The index of racial disparity decreased from 51.4% to 22.6% between 2005 and 2014 for receipt of breast reconstruction [48]. When examining a different database, the overall incidence of reconstruction remained at 48%, with 58% for White and 34% for Black patients in 2018 [49]. In a large study of the National Cancer Database between 2004 and 2017, American Indian/Alaska Native rates increased, ranging from 13 to 47%, and were independently associated with a decreased likelihood of reconstruction compared to Whites [12].

Significantly, Black patients underwent 3.6 times less reconstruction than White patients when controlling for age, insurance type, income, comorbidities, age, and insurance status [42]. During the years 2004–2016, Black patients received less reconstruction when broken down between each age group compared to their White counterparts [29]. African Americans were 30% less likely to receive breast reconstruction compared to Caucasians at a comprehensive cancer center during 2000–2012 [50]. Black women also experienced the longest interval to treatment [51].

When adjusted for household income, African American (AA) and Asian women had a lower adjusted rate of immediate breast reconstruction compared with White women; however, Middle Eastern and Hispanic women did not. Middle Eastern women had a higher rate of delayed breast reconstruction compared with other ethnic groups [47]. 

All minorities remained at lower rates of immediate breast reconstruction compared to Whites in multiple national databases and multi-institutional studies [15,52]. African Americans were the least likely to receive immediate breast reconstruction. Middle Eastern, Hispanic, and Asian women were less likely to receive immediate or delayed reconstruction compared to White women, and African American women were the least likely to receive immediate breast reconstruction [47]. In a single-state database between 1998 and 2006, the rates of IBR increased to 73% for White patients, 32% for Black patients, and 38% for Hispanic patients, finding that the odds of being African American or Hispanic reduced immediate reconstruction by 43% and 26%, respectively [33]. From 2005 to 2011, rates of immediate reconstruction for races were Caucasian 35.2%, AA 24.6%, and Latina 33% [32]. One study found that Black patients were less likely to receive IBR compared to Whites between 2012 and 2017, but by 2021, both races had similar odds of IBR [53]. There are instances of single institutions finding no racial differences among breast reconstruction rates on univariate analysis and that they were comparable to national rates [8,41]. No indication of racial differences in reconstruction were found in the Department of Defense database between 1998 and 2007 or when stratified by age [30].

In a study of the NCDB database and Medicaid expansion, reconstruction was more common among white women in both pre and post-Medicaid expansion time points compared to Black patients and even less so for Asian women [54]. There was a decrease in the disparity in 2014 in the early-expansion states and again in 2017 in late-expansion states [31]. White women were also more likely to receive immediate reconstruction if they had private insurance compared to public insurance, and Asian patients were the least likely to have reconstruction in all insurance-type categories [39].

When a higher density of plastic surgeons was present, all racial subgroups improved their rates of immediate reconstruction, although disparity among races remained with Caucasian 59%, Hispanic 47%, AA 42%, and APINA 41% [23]. There were racial differences among complication rates as well: Black (20.4%) patients relative to White (17%), Hispanic (17.9%), and Asian (13.2%) patients were more likely to have breast reconstruction complications [19].

### 3.3. Age Disparities

Younger patients (<60) undergo breast reconstruction at a statistically significant higher rate [2,8,17,22,30,34,36,39,40,45,55,56]. In a study of the NCDB between 1998 and 2007, younger age was the most significant predictor of reconstruction. Patients under age 50 were 3.72 times more likely to undergo breast reconstruction than those aged 50 and over in 1998–2000, which narrowed over time to 3.40 in 2005–2007 [10,52].

In reviewing multiple large databases, younger patients also had higher rates of immediate breast reconstruction [35] and were more likely to receive immediate breast reconstruction at an average age of 46.7 +/− 9.9 years versus ages 62.9 +/− 12.8 years who did not undergo reconstruction [57]. Age < 45 received immediate reconstruction at a rate of 52.9% during 2005–2011 versus 40.1% in ages 45–65 and 11.1% in the age group and >65 [32]. In a large, single-state study during 2005–2015, all age groups had a significant increase in immediate breast reconstruction [33].

Ages 30–39 were more likely to receive implant reconstruction than ages 40–49, while ages 50–59 were found to have more autologous and delayed reconstruction compared to age group 40–49 [14]. In a national study on Medicaid expansion, reconstruction was more common among women under age 50 at 35.1% before expansion and 46.3% after expansion versus >60 at 16.7% before expansion and 22% after expansion [54]. In a small study of a single institution, reconstruction was more likely in the age group 40–49 for patients undergoing microsurgical breast reconstruction [58].

### 3.4. Insurance Disparities

The privately insured had higher rates of breast reconstruction than those with public insurance [11,17,23,33,34,36,37,42,59], compared to Medicaid [37,55] and Medicare [37]. Reconstruction receipt was highest amongst Tricare prime beneficiaries [30]. In a study of NIS during 2002–2006, it was found that women with private insurance coverage received reconstruction at a rate two times higher compared to those with public or uninsured coverage (48.8%, 23%, and 18%) [39]. Privately insured patients received breast reconstruction at 60% and those with public insurance at 20% regardless of race and ethnicity [23], including more reconstruction for all types, both autologous [14] and immediate [35].

Medicaid expansion heavily influenced the rates of reconstruction for the better across all races and insurance types but not equally among the groups. In a large national study during 2004–2016, Medicaid expansion was associated with breast reconstruction for African American but not Caucasian women [29]. A disparity amongst race was found with immediate breast construction in a national multicenter study, with Caucasians receiving 34%, Hispanic 28%, AA 24%, and APINA 24% [23].

Those who were publicly insured were more likely to receive higher rates of IBR if there was a higher density of surgeons in their geographical location [23]. In a national multicenter study, for those with private insurance and living in a highly dense plastic surgeon area, there remained a disparity between races. Caucasian women had higher IBR at 84%, Hispanic 65%, AA 60%, and APINA 58% [23]. Moreover, 25.5% of privately insured patients underwent IBR compared to 17.7% of those with Medicare insurance. The majority of private patients underwent IBR in the Northeast (29.9%) compared to most Medicare patients in the South (30.9%) There was a trend increase from 3.9% in 1992 to 47.2% in 2013 in privately insured patients, while there was an increase from 2.3% in 1992 to 43.7% in 2013 in Medicare patients [60].

In a 10-year study from 2005 to 2015 regarding all payer types for IBR, Medicaid saw the largest increase in the years 2010–2011, Medicare in 2012–2013, and private insurance in 2008–2009. There was only a significant increase in autologous reconstruction for Medicare and commercial insurance, with no changes for Medicaid [34]. When examining the later years, in 2017, the post-mastectomy reconstruction rate of private payers was 60.6%, while that for Medicaid and the uninsured were 39.1% and 34.6%, respectively. In the states that underwent Medicaid expansion, there was a higher rate of overall reconstruction (38.5%) compared to non-expansion states (28.7%) [54]. However, patients who had Medicare were 21.4% less likely to receive reconstruction, which continued throughout the years analyzed even with increased Medicaid coverage [54]. Reconstruction was found to be more common among women living in expansion states compared to non-expansion states [54]. Women in an urban location with private insurance received reconstruction at a rate two times higher compared to public or uninsured patients. In a study of the NIS database between 2002 and 2006, the odds of reconstruction were not significantly different for race and ethnicity [39]. Medicaid expansion was shown to significantly increase breast reconstruction rates [61], particularly among private payer insurance [10,43]. While there was a 6.8% increase in reconstruction procedures after Medicaid expansion, there was an overall 9.6% increase in all treatment coverage by Medicaid, so the demand for breast reconstruction was not met [62].

### 3.5. Geographical Disparities

Many studies reported varying trends in the areas receiving the most reconstruction. One reported the Northeast was less likely [14], while some reported higher rates [29,36,54]. One large national study found that AA in the Northeast and West had a higher association with reconstruction than their Caucasian counterparts [29], with AA in the southern states having the lowest rates. A universal trend was increasing population density is positively associated with reconstruction [45]. A highly dense population, a metropolitan, or urban location had an increased likelihood of all post-mastectomy breast reconstruction [2,5,40,59] and immediate breast reconstruction specifically [60]. An urban location was more associated with reconstruction regardless of race; however, it was less associated with African Americans versus Caucasians [29]. The individual rates of reconstruction for urban, near-metro, and rural areas were 31.1%, 20.4%, and 13.4% and were highly predictive of reconstruction when all other variables were controlled for in a single-cancer-center registry. Rural-area patients were 64% less likely to undergo reconstruction versus near-metro patients, who were 46% less likely than urban counterparts [43,44].

In a study of NIS during 2002–2006, Shippee et al. [39] found that women in an urban location had more than three times the rate of reconstruction in rural hospitals: 44.1% vs. 14.3%. Urban location and teaching status correlated greater odds of reconstruction amongst all payer types except for in the urban setting, which did not demonstrate increased odds in the uninsured. Odds of BR were lower for women with public insurance in the West compared to the Northeast, although this was not a statistically significant association [39].

In a large healthcare utilization cost study from 2012 to 2019, the majority of rural patients underwent reconstruction at urban teaching hospitals, while rural patients were more likely than non-rural patients to have surgery at a rural hospital (6.8% vs. 0.7%). The cost of autologous breast reconstruction was higher at urban compared to rural hospitals (USD 30,066 urban vs. USD 25,049 rural) [3].

Those who underwent reconstruction lived in ZIP codes that received approximately USD 2000 more in annual income compared to patients who did not undergo reconstruction [63]. Living in an area with a higher-than-average education was also correlated to increased odds of receiving reconstruction [11].

There was, however, a regional variation in immediate breast reconstruction showing there were pattern differences in smaller geographical regions [24]. In a large statewide database, increasing AA composition of the patient’s neighborhood had a statistically significant negative association, and it was found that overall, 47% of AA were less likely to receive IBR [45].

### 3.6. Travel Disparities

Patients with Medicaid insurance and living at an increased distance from an accredited breast surgery center were 32% less likely to receive IBR, while non-Medicaid patients residing closer to an accredited breast surgery center had a predicted reconstruction probability rate of 50% [64].

### 3.7. Income Disparities

An income of >USD 65,000 has been associated with greater rates of BR [2,11,34,36,45,59] LeBlanc et al. [54] and increased likelihood of receiving implant reconstruction [14]. Those in the top income quartiles between 2004 and 2018 had an increased likelihood of undergoing immediate breast reconstruction [10,26,65]. In 2007, patients living with a median household income of >USD 46,000 were 1.66 times more likely to undergo breast reconstruction [10]. Living with a median household income of >USD 75,000 increased the odds of having immediate breast reconstruction by 1.50 times [33].

### 3.8. Individual and Behavioral Disparities

When examining additional personal factors, a small study cited that married women pursued immediate reconstruction more often than single patients (13% vs. 9%) [46]. Minority women expressed a greater desire for additional information about reconstruction options compared to White patients [21]. Hispanic women who underwent reconstruction did so with an interest in restoring “normalcy” and achieving a more “feminine” appearance by returning to how they felt about their bodies before their surgeries, as cited in a survey. Overall, the leading concerns for undergoing reconstruction were safety, avoidance of additional procedures, or aesthetics. They reported a heavy mental and psychological burden as well as exhaustion and being overwhelmed by the process, although the majority had support within their community [20]. Between 1995 and 2004, having a computer in the house was not significantly associated with an increased rate of immediate breast reconstruction [45].

### 3.9. Education Disparities

Higher education level has been positively correlated with rates of reconstruction [36,54], while Mahmoudi showed that living in a neighborhood with a greater proportion of college-educated residents increased immediate breast reconstruction odds by 4.43 [33].

### 3.10. Language Disparities

Language has been identified as a barrier to BR access in a few small studies. [49] uncovered that the need for an interpreter was associated with reduced rates of discussion about plastic surgery options and reconstruction procedures within clinic visits [49]; similarly, patients who spoke a language other than English as their primary language experienced lower odds of being informed about breast reconstruction options [66]. However, in a small, single-institutional safety-net hospital, no variation was found between English and non-English-speaking patients [41].

### 3.11. Hospital Type Disparities

Hospital systems and insurance status affect rates of breast reconstruction. Academic, research, and teaching hospitals [35]; Integrated Network Cancer Programs; and large hospitals [60] demonstrated higher rates of breast reconstruction compared to community cancer programs [36]. Being treated at more than one cancer center also increased the likelihood of receiving breast reconstruction [36]. Odds of reconstruction were lower at non-teaching and cooperative oncology group hospitals and lower-volume facilities, even when controlled for race and ethnicity (Onega et al. [5]). In an analysis of the NIS database from 2002 to 2006, Shippee et al. [39] found that less than one-third of women in non-teaching hospitals underwent reconstruction compared to about one-half at a teaching hospital [39]. Data from the NCB between 2004 and 2015 supported this inconsistency in treatment by hospital type, with 43.7% of patients treated with BR at an academic center versus 32.5% at a community hospital [67].

Examining the NCB between 2004 and 2015 found that patients treated at an academic center were younger than those treated at a non-academic center and traveled further distances for their treatment [67]. Academic centers were more likely to treat younger patients and provide breast reconstruction regardless of insurance status. Uninsured patients received reconstruction at a rate of 24% at an academic center compared to 19% at a community hospital. Patients under the age of 40 received reconstruction at an academic center at a rate of 66% versus 58% at a community hospital, a trend consistent across age all ranges [67]. Immediate breast reconstruction was also less likely to be performed at community hospitals (34.41%) compared to academic hospitals (53.24%) [34].

In a large urban city with both a private cancer center and public hospital, the rate of reconstruction at a private cancer center was 56.6% compared to 36.2% at the nearby public hospital. Immediate breast reconstruction with private insurance at the private cancer center was 93.6% versus 40.8% private insurance at the public hospital. When a patient had public insurance and received care at the private cancer center, the rate of reconstruction was 86.7% versus 45.5% at the public hospital. Simply receiving care at a private institution resulted in 22.96 times greater odds of undergoing reconstruction for non-private insurance patients compared to treatment at a public hospital [68]. Between 1998 and 2007, 30.4% of patients were treated in an academic or research hospital. This increased to 31.5% between 2005 and 2007 [10].

A surgical desert is classified as six or fewer general surgeons per 100,000 people in an underserved or rural county. In California, during 2007–2010, surgical deserts had a 14% rate of breast reconstruction compared to 39% in non-surgical deserts [69].

### 3.12. Healthcare System Disparities

Notable publications examined within this systematic review have highlighted the implications of patient–clinician communication on surgery decision making and patient referral patterns. Overall, 28% of mastectomy patients who did not receive IBR declined a referral to plastic surgery, 16% were seen in the clinic by plastic surgery but declined reconstruction, and 2% were referred but did not appear at the appointment. The other 4% had unknown reasons for foregoing surgery. The other 50% received delayed reconstruction or were not candidates. The work of [70] supported this healthcare complex dynamic, showing that between 2013 and 2018, patients did not undergo reconstruction due to declining a referral by choice (23%) or meeting with a plastic surgeon but not undergoing reconstruction (25%) [26].

Reconstruction is discussed less frequently with patients who are older, less educated, uninsured, or part-time workers or whose primary language is not English [13]. One institution found that in a 5-year period, only 12.8% of mastectomy patients were referred to plastic surgeons [8]. Minority women were less likely than White patients to be assessed by a plastic surgeon before their initial surgery [21]. Another single-institutional study recorded that 64.8% of BR patients were seen and evaluated by a plastic surgeon before their first operation. For those not evaluated preoperatively, patient preference (48%) or ineligibility for IBR (26%) were cited as the rationale. Only 52.1% would eventually undergo reconstruction (83.8% were immediate, 19.4% autologous, 74.2% implant-based, and 1.6% combination), and if the timing of reconstruction was delayed, 66.7% received autologous reconstruction after mastectomy [41].

In a chart review of clinical communications, 70.7% of all patients who had a documented discussion on reconstruction received reconstruction. Overall, 65% underwent reconstruction if they had documentation at one visit, 70% at two visits, and 85% ≥ 3 visits. The greatest predictor of reconstruction surgery in this study documented the record of the preoperative discussion. Independent of documented communications, lower rates of reconstruction were associated with increasing age, lower education level, race, birthplace outside of the U.S., and a primary language other than English [13].

Based on patient survey responses within a single-institution evaluation, females electing for microsurgical BR and those who were consulted by plastic surgeons were more likely to be self-referred or referred by another plastic surgeon. The internet was emphasized as the primary patient educational tool for 85% of microsurgery patients and 50% of non-microsurgical breast reconstruction patients [58].

Studies conducted in a single-institution setting did not attribute race, lower household income, or government insurance status to significant differences in plastic surgery referral patterns, rates of reconstruction, or IBR utilization [70]. In contrast, in the context of a state-wide multi-institutional study, AA race, charge to insurance, and provider independently predicted the type of reconstruction procedure performed, with race being the most clinically significant predictor of autologous-based reconstruction [16]. This study also found a disparity in charges for reconstruction. AA were more likely to receive a greater charge to their insurance company versus their White counterparts (USD 5580 vs. USD 3990) [16]. These findings were corroborated by differences observed in surgical oncologist referrals for breast reconstruction by [47], along with differential rates of patient referral acceptance, plastic surgeon recommendations, and the choice disparity between races when considering surgical reconstruction.

After the implementation of federal mandates to cover breast reconstruction in 1998, there was an increase of 31–36% in breast reconstruction utilization nationally [7]. Immediate breast reconstruction increased across all insurance types and races. Medicaid saw a 4.2-fold increase, Medicare 2.9, private 2.6, and self-pay 2.1 [65]. White, Black, and Asian patients all experienced increased frequency of IBR; however, after controlling for confounders, only White patients remained more likely to undergo IBR after policy enactment [71]. The 2011 New York Public Health Law 2803-o, requiring physician communication about breast reconstruction in all patients undergoing mastectomy in the state of New York, led to more equitable discussions about BR for White and minority patients. Of concern, no concordant normalization of IBR rate disparity was observed between White and AA individuals [72].

### 3.13. Surgeon Disparities

Individual surgeon demographics have proven to influence reconstruction rate disparities. In a small cohort comparing female to male breast surgeons, female surgeons had 3.7 times greater odds of operating on patients who underwent reconstruction after controlling for age, race, insurance, and type of surgery [55]. Patients cared for by female breast surgeons were also more likely to undergo breast reconstruction, whereas if reconstruction was completed by a hospital-affiliated plastic surgeon, patients were more likely to receive autologous reconstruction compared with implant-based [22]. Surgeons operating on a higher case volume of breast cancer patients, specified as >51 procedures per year, were more likely to perform BR surgery [66].

When having physicians undergo implicit bias testing, no bias was found among physicians and disparities in breast reconstruction rates, complications, or cost. However, there remained a difference in utilization ratio between race and ethnicity [73]. There was no correlation between immediate breast reconstruction rates and surgeon graduation year in a single institutional study [57].

Between 2017 and 2018, a single institution demonstrated that 68% of patients had a discussion on plastic surgery options with their breast surgeons, and 62% of patients received referrals [21]. If a patient was older or had non-private insurance, their chances of preoperative discussions and referrals decreased, yet no significant disparity with race or ethnicity was observed on univariate analysis. After multivariate adjustment, however, significantly disparate reconstruction rates were associated with Black race and BMI ≥ 35. Of note, BMI alone did not lower breast reconstruction rates in Black versus White women. While there were equal rates of plastic surgery discussions and referrals, Black women experienced lower rates of referrals for reconstruction compared to White women [49]. Another distinctive study examining race and patient communication highlighted within a single-institutional study population lower rates of referrals for African Americans; African Americans were also subsequently less likely to accept offered referrals, consent for reconstruction, and select implant reconstruction if offered [47].

The average total charge reimbursement at a single institution for implant-based reconstructions was 16.3% via Medicaid, 28.3% via Medicare, and 67.2% via private insurance. The average annual autologous reimbursement rate was 12.37% for Medicaid, 22.9% for Medicare, and 35.35% for private insurance. The hourly reimbursement estimates for Medicaid patients receiving autologous reconstruction were the lowest on average, while the highest rates were observed in privately insured patients undergoing implant-based reconstruction. One study’s results showed a steady decline in reimbursement rates for autologous reconstruction for all payer types and narrowed variability amongst different groups for implant-based reimbursements [74].

In a qualitative survey study of 10 breast/general surgeons and 5 plastic surgeons from the NIS database in Wisconsin, the surgeons recognized their significant control over which patients undergo reconstruction. Both groups of surgeons stated that general/breast surgeons are perceived as “gatekeepers” to reconstruction and who received a referral to plastic surgery. They reported “how” and “when” reconstruction was presented and discussed during clinical consultations. Both groups recognized the importance of reconstruction, even with an attributed increased risk of multiple operations. General/breast surgeons explored the consideration of the benefits and limitations of the patient and acknowledged a physician’s internal bias influences patient interest in reconstruction. The surgeons acknowledged personal bias in believing repeated procedures and clinic visits required a “motivated woman” for reconstruction, and not all women would prioritize it. They cited the disproportionate burden caused by work leave, travel expenses and logistics, and out-of-pocket financial costs as probable limitations for socioeconomically disadvantaged patients. Early education, financial discussions, and a decreased travel burden would improve access for their patients and be presented as the most important alleviating factors for patients of lesser socioeconomic means [75].

### 3.14. Plastic Surgeon Density

When plastic surgeon density was 2.2 per 100,000 people, the relationship to breast reconstruction had a correlation coefficient of 0.69. In areas with greater than three plastic surgeons to 100,000 people the reconstruction rate was 60%, while if the prevalence of surgeons was less than one plastic surgeon per 100,000 people, the rate fell to 23% [23]. In a single-state study comparing Appalachia (rural) versus non-Appalachia territory (urban), the rate was 1 versus 20 plastic surgeons counted in these areas [44]. A lower number of plastic surgeons within a 50-mile radius was associated with a decreased rate of reconstruction, as found in a state cancer registry [2].

## 4. Discussion

This systematic review is a comprehensive outline of sociodemographic and economic factors affecting access to breast reconstruction. While there has been an overall increase in rates of breast reconstruction nationally and within most demographics in the last 20 years, disparities between groups remain significant. Minority groups, public-insured or uninsured, rural, low income, access to community hospital, and non-English speaking are the many known and studied contributing variables, although there remain many unknown factors or areas with limited data.

Several previous systematic reviews have suggested solutions such as patient education, bias training, legislation [18], cultural competency training to reduce provider bias [18], automatic referrals to plastic surgeons, and patient education [76] to mitigate barriers in minority groups. In an extensively outlined framework for identifying and addressing racial disparities in addition to the above listed options, Doren et al. (2023) [6] included recommendations for infrastructure and restructuring care centers.

From 1998 to 2015, almost all rates of autologous, immediate, and delayed breast reconstruction doubled [10,31,35]. Multiple papers examined the rate of change pre- and post-federal mandate enactment and cited this as a factor that contributed to the positive increase. Additional considerations to continue increasing rates is to gauge patient and provider awareness of the federal mandate.

Between 2005 and 2015, rates for implant-based reconstruction ranged from 75.7 to 78% and 22 to 37% for autologous reconstruction [32,34]. Autologous reconstruction remained more often delayed [14]. Those who received implant reconstruction were more likely to be in the age groups 30–49, Caucasian and Asian, higher income, cared for in a larger hospital, and within the Northeast region of the United States [14]. Immediate reconstruction rates varied based on databases but reached 59% in 2015 [35].

Minorities continue to receive less reconstruction overall compared to White/Caucasians [37], with American Indian/Alaska Natives [12], Middle Eastern [47], Hispanic [46], and Black [11] patients receiving the least amount even while controlling for socioeconomic status, insurance type, age, and geographical variables [42]. In 2018, the overall rate was 48%, with 58% for White and 34% for Black patients [49], who were 3.6 times less likely to undergo reconstruction while controlling for all variables [42]. Black patients also had the longest interval to treatment [51]. The index of racial disparity decreased by more than half in 2014 [48], and all minorities have seen an overall increase in rates despite the existing disparity. Race was predictive of reconstruction type for Non-Hispanic Black and Hispanic women, who were more likely to receive autologous [16] versus implant-based reconstruction [46], which was higher in Caucasian and Asian ethnicities [14]. However, one study found that Black patients were less likely to receive IBR compared to White patients between 2012 and 2017, but by 2021, both races had similar odds of IBR [53], and many single-institutional studies did not find racial differences in reconstruction [30] or had rates that were comparable to national numbers [8,41]. This underscores the importance of looking deeper into the causes of racial disparities, most likely found within quantitative data that have not been examined or qualitatively through surveys and understanding cultural differences.

Younger age was the most significant predictor of reconstruction [40,52], as these patients had higher rates of IBR [35] and were more likely to receive implant reconstruction [14]. Multiple factors can contribute to age disparities. One includes provider bias, but additionally, insurance likely plays a large role. Medicare for patients 65 and older consistently showed decreased reconstruction rates [37]. Those with private insurance received reconstruction at a rate two times higher than patients with public insurance [39] regardless of other variables and for all reconstruction types [23]. An income of >USD 75,000 or in the top income quartiles [10,26,65] was also significantly associated with higher rates and implant reconstruction [14]. In 2017, the rates for private insurance were 60.6%, Medicaid 39.1%, and uninsured 34.6%.

After Medicaid expansion, overall rates increased for all insurance types, although minority races still received less reconstruction [23]. One mitigation of reconstruction rates was plastic surgeon density [19,23] and urban location, which increased immediate breast reconstruction, particularly in those with private insurance [39], regardless of race [29]. Rural areas are the least likely to receive reconstruction compared to their urban cohorts [43,44], which remained the same across all the years examined. An increase in access to plastic surgeons could help address rural access; however, this is a specialty currently found more commonly in more populated areas. An addition of training programs or expansion in providers of breast reconstruction would be required.

Large urban teaching hospitals and treatment at a cancer center yielded an increased likelihood of receiving immediate breast reconstruction, with 22.96 increased odds of undergoing reconstruction if receiving care at a private center compared to a public hospital [68], particularly if the patient had private insurance. Living in ZIP codes with higher annual income [63] and higher-than-average education [11] were also notable characteristics that led to increased rates. Higher education of a patient [36], English as the primary language [66], and decreased distance to accredited breast centers were all associated with increased rates of breast reconstruction as well, further supporting the finding that those who have access to socioeconomic means will benefit from reconstruction.

Provider behaviors were also contributing factors affecting rates. While there were limited studies on the insight of a surgeon regarding referrals, one survey on both general/breast surgeons and plastic surgeons stated that general/breast surgeons were the gatekeepers of who would receive a referral for reconstruction [75]. There was a difference in utilization between race and ethnicity [73], increased surgeries if the provider was female [55], and higher yearly case volumes [66]. A single institution reported that 68% of patients had a discussion about plastic surgery options with their breast surgeons, and 62% of patients received referrals, and the statistics decreased if a patient was older or with public insurance [21]. African American women were also less likely to receive or accept referrals or consent to undergo reconstruction if offered [47]. Many provider characteristics are relevant when considering which demographic receives breast reconstruction, particularly the conditions under which referrals are given, but this is a highly understudied area.

One notably interesting factor not frequently examined in the literature is reimbursement. At a single institution, the hourly reimbursement rates were the lowest for Medicaid patients receiving autologous reconstruction and highest in privately insured patients receiving implant-based reconstruction [74]. African Americans were also more likely to have their insurance charged at a higher price compared to their White counterparts [16]. When considering reimbursement and surgeon recommendations, there is strong evidence that patients who receive a larger monetary compensation may be provided with reconstruction more often. A greater understanding of financial influences and monetary considerations would help isolate areas that could be reformed to enable greater access.

### 4.1. Future Considerations

Trends in the data show a slowing of reporting in more recent years, with the bulk of data published around the federal mandate enactments, highlighting their improvement in helping more women obtain reconstruction. However, there remain limited data beyond the commonly reported studies and a lack of examination of the additional contributing factors.

One of the major factors associated with increased rates is the presence of plastic surgeons; therefore, the need for further understanding of how their presence interplays with the current system is crucial. A wider understanding of their cultural competency training, attitude, or bias will help elucidate further areas of improvement. Most importantly, reimbursement data for providers, hospitals, and insurance types will surely give an even greater understanding of the system governing reconstruction rates.

With the recent global changes of the past four years and the growth of the consumer knowledge base due to social media and the public’s understanding of plastic surgery, an update in trends and patient awareness is worth examining to expand on patient education. These include patient understanding of insurance coverage, out-of-pocket costs, and reconstruction options available to them.

The current data reported are robust; however, many of the variables are patient demographic factors, which cannot be changed. Therefore, future research needs to isolate and identify factors that could be adjusted to offer greater solutions and an increase in access.

### 4.2. Limitations

This systematic review was not without limitations. The literature was restricted to the English language and that published within the United States, potentially excluding additional work and leading to language bias and findings applicable only to the United States. Although a comprehensive search of multiple databases was carried out, publication bias may be present due to the presence of positive results, which are more likely to be published. Our inclusion criteria, although vast, may also have potentially excluded relevant studies. Variations in methodological assessment of quality across the literature may have influenced the overall validity of the conclusions.

## 5. Conclusions

Federal mandate enactments have improved access for women undergoing postmastectomy breast cancer reconstruction, particularly for younger, White, privately insured, urban-located patients. The studies that have been published more recently have a stronger focus on disparities, particularly among races, and show a continued disparity for minorities, patients with lower income, those in rural communities, and public insurance payers. Provider data are limited, but the available analyses shed light on their apparent influence on reconstruction types and rates. Additional understanding of providers, hospitals, and insurance payers may offer additional insights into ways to overcome barriers to post-mastectomy breast reconstruction.

## Figures and Tables

**Table 1 medicina-60-01169-t001:** Disparity categories and frequency of discussion within the literature.

Study Category	Number and Percentage of Topic Discussed (*n* = 70)
Sociodemographic	61 (87.1%)
Insurance	38 (54.2%)
Geography	21 (30%)
Hospital type	13 (18.6%)
Income	12 (17.1)
Surgeon	9 (12.86%)
Education level of patient	5 (7.14%)
Communication	5 (7.14%)
Federal mandates	5 (7.14%)
Patient satisfaction	1 (1.4%)
Decision making	1 (1.4%)
Reimbursement	1 (1.4%)
Language	1 (1.4%)

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
