# Peer review of "Post-Mastectomy Breast Reconstruction Disparities: A Systematic Review of Sociodemographic and Economic Barriers"

_medicina, 2024, doi:10.3390/medicina60071169_

Round 1

Reviewer 1 Report (Previous Reviewer 2)

Comments and Suggestions for Authors

This is a systematic Review of Sociodemographic and Economic Barriers for Post-Mastectomy Breast Reconstruction. It addresses the Sociodemographic and Economic Barriers for breast reconstruction which is an important topic. 

Similar topics have been done previously but this paper provided an update.

It also provides an update of the Sociodemographic and Economic Barriers for breast reconstruction, however, this is only applicable to patients in the United States.

Authors can consider exploring the Sociodemographic and Economic Barriers to breast reconstruction globally rather than restricting it to United States.

Conclusions are consistent. All main questions addressed. References, tables and figures are appropriate.

Author Response

Comment 1:  "It also provides an update of the Sociodemographic and Economic Barriers for breast reconstruction, however, this is only applicable to patients in the United States.

Authors can consider exploring the Sociodemographic and Economic Barriers to breast reconstruction globally rather than restricting it to United States."

Response 1: Thank you for your review and we agree it is focused on the United States. Due to the global variation in insurance systems, access to medical care, and sociodemographic influence, the addition of global barriers would stretch beyond the interest of this paper. It is a topic that deserves its own review and will be considered for future research.

Reviewer 2 Report (New Reviewer)

Comments and Suggestions for Authors

Authors should be congratulated for their effort to present this systematic review focusing on clarifying the impact of several socio-economic disparities in post-mastectomy Breast Reconstruction (ΒR).

Considering the manuscript their Introduction is adequate, well-organized providing readers with valuable information, explaining the rational of their study. Their aim is well- presented in the last sentence of the last paragraph (lines 78-80).

In the subsequent section, authors thoroughly explained every step of their methodology. There were provided details concerning selection criteria, search term conditions, sample screening. Data collection completed this section.

Results stressed the increased rate of BR in all ethnic groups throughout the years. However, all minorities showed lower rates of Immediate Brest Reconstruction (IBR) compared to white patients. Interestingly, white women disposing private insurance had more often IBR compared to them with public insurance. Younger patients undergo BR at a statistical significant higher rate compared to other age groups. A disparity observed in the type of reconstruction. Specifically, in younger ages (patients 30-39 years) implants was the preferred reconstructive method. Autologous reconstruction was the favorite method of the next age group (40-49 years). Type of insurance (private vs. public) was another factor, which predicts the rate of undergoing reconstruction. Essential was the role of higher density of surgeons in the higher rates of IBR, while all type of insurance demonstrated a large increase in the rates of IBR. Geography has a particular role in the BR. Subsequently, increased population density characterized by a higher reconstructive rate. Patients from urban areas were prone to IBR compared to them from rural areas. Interestingly, the majority of rural patients preferred urban teaching hospital for their BR. Educational level was another interesting factor, which favored BR., as was the case of the annual income. Academic centers attracted younger patients for their BR regardless of the insurance level. Of note the introduction of the term of “surgical desert” for areas with less of 6 general surgeons/ 1000000 people. Individual surgeon demographics apparently influenced rates of reconstructive disparities. Female breast surgeons played a crucial role in patients’ decision to undergo BR. Finally, plastic surgeons’ density was important for the rate of BR in a certain area.

In the ensuing discussion section authors highlighted all the important issues of their results, ideally completing a well-written article that is suitable for publication.

Author Response

Comments 1: In the ensuing discussion section authors highlighted all the important issues of their results, ideally completing a well-written article that is suitable for publication.

Response 1: We appreciate the thorough review of this manuscript, thank you.

Reviewer 3 Report (New Reviewer)

Comments and Suggestions for Authors

The manuscript “Post-Mastectomy Breast Reconstruction Disparities: A Systematic Review of Sociodemographic and Economic Barriers” is extensive and well written and I appreciate the authors for the research approach they have taken. I thoroughly enjoyed reviewing this manuscript and only have some minor requests for revision.

1.    The results are explained in detail and the readers might find it exhaustive. The suggestion would be to shorten the paragraphs under each subtitle.

2.    If tables could be included with the percentage and sociodemographic details, extensive writings could be reduced. A single paragraph explaining each of the titles under the result section could be included along with a table to improve the readability of the paper.

3.    The discussion section could be reduced.

4.    Line 343 and 351 – Check ‘recon’

Author Response

Comment 1: The results are explained in detail and the readers might find it exhaustive. The suggestion would be to shorten the paragraphs under each subtitle.

Response 1: The paper is exhaustive, however, due to the nature of the issue we determined that all points chosen to be included were necessary for discussion.

Comment 2: If tables could be included with the percentage and sociodemographic details, extensive writings could be reduced. A single paragraph explaining each of the titles under the result section could be included along with a table to improve the readability of the paper.

Response 2: While readability is of the utmost importance, we believe that removing any discussion points would be a disservice to the topic.

Comment 3: The discussion section could be reduced.

Response 3: During the writing of the manuscript we determined everything that was included was required. Shortening the discussion may lead to a misunderstanding of the contributing factors to breast reconstruction access issues.

Comment 4: Line 343 and 351 – Check ‘recon’

Response 4: "Recon" was found and edited to "reconstruction"

This manuscript is a resubmission of an earlier submission. The following is a list of the peer review reports and author responses from that submission.

Round 1

Reviewer 1 Report

Comments and Suggestions for Authors

This systematic review entitled "Post-Mastectomy Breast Reconstruction Disparities: A Systematic Review of Sociodemographic and Economic Barriers" performed by Vangsness et al. evaluates the sociodemographic and economic barriers to post-mastectomy breast reconstruction (BR), revealing persistent disparities despite overall national increases in BR rates.

While legislative measures have improved access, minority groups, individuals with public or no insurance, those from rural or low-income backgrounds, non-English speakers, and patients treated at community hospitals continue to face significant barriers to BR.

The review identifies multiple factors influencing these disparities, including race/ethnicity, insurance status, geographic location, income, hospital type, and patient education.

The study is surely topical and impressive in its size; however, I have some major concerns that need to be addressed before potential publication:

- In the Introduction section, page 15, lines 89,90, the authors state that one of their aims is to "improve health equity among the breast cancer population.". However, how they intend to do that? Please clarify and specify in the Introduction section.

- Table 1 is too long and it should be condensed a lot;

- In the Methods sections, the authors state that one of the search query was "rate of breast reconstruction after lumpectomy". However, this is profoundly misleading with the title and scope of the systematic review. Lumpectomy is usually considered a breast-conserving surgery, and there is no breast reconstruction after that (sometimes just some remodeling). Therefore, either the research design is highly flawed, or the authors did not use a correct expression for their reserch query. Please clarify and modify;

- In the Methods section, the authors should add a paragraph of definitions. How was breast reconstruction defined? How was immediate breast reconstruction defined? How was delayed breast reconstruction defined? Implant-based or autologous reconstruction? There is a huge difference;

- The Results section should be better structured. Try to intersperce paragraphs with Tables detailing the results; 

- In the Discussion section, the authors should mention that, breast surgeons should do everything in their power to maintaing the original breast and prefer a breast conserving surgery for breast cancer over mastectomy, if technically possible. This is particularly true for better oncological outcomes (please cite, PMID: 37600290) and also in cases of second breast conserving surgeries after ipsilateral breast cancer rucurrences (please cite, PMID: 33431329).

- What are the solutions for these disparities proposed by the authors of the systematic review? Try to be more specific in your conclusions.

Author Response

- In the Introduction section, page 15, lines 89,90, the authors state that one of their aims is to "improve health equity among the breast cancer population.". However, how they intend to do that? Please clarify and specify in the Introduction section.

  1. This was clarified by the adjustment: “This systematic review aims to comprehensively outline and specify sociodemographic, economic, and clinical factors contributing to the distribution of breast reconstruction for cancer patients.”

- Table 1 is too long and it should be condensed a lot;

  1. This was moved to supplemental

- In the Methods sections, the authors state that one of the search query was "rate of breast reconstruction after lumpectomy". However, this is profoundly misleading with the title and scope of the systematic review. Lumpectomy is usually considered a breast-conserving surgery, and there is no breast reconstruction after that (sometimes just some remodeling). Therefore, either the research design is highly flawed, or the authors did not use a correct expression for their reserch query. Please clarify and modify;

  1. Lumpectomy is a well-accepted technical term used to describe breast-conserving therapy.

- In the Methods section, the authors should add a paragraph of definitions. How was breast reconstruction defined? How was immediate breast reconstruction defined? How was delayed breast reconstruction defined? Implant-based or autologous reconstruction? There is a huge difference;

  1. A paragraph of definition was included within the methods section to define the search terms

- The Results section should be better structured. Try to intersperce paragraphs with Tables detailing the results; 

  1. The results section was structured in order to separate the ideas

- In the Discussion section, the authors should mention that, breast surgeons should do everything in their power to maintaing the original breast and prefer a breast conserving surgery for breast cancer over mastectomy, if technically possible. This is particularly true for better oncological outcomes (please cite, PMID: 37600290) and also in cases of second breast conserving surgeries after ipsilateral breast cancer rucurrences (please cite, PMID: 33431329).

  1. This was not included in the discussion, as conserving the original breast is not always ideal or preferred, it is variable for each patient. 

- What are the solutions for these disparities proposed by the authors of the systematic review? Try to be more specific in your conclusions.

  1. The solutions discussed include a broader understanding of physician awareness, qualitative data on cultural differences of patients and location, increasing in training programs to varying geographical locations, and an understanding of monetary influences.

Reviewer 2 Report

Comments and Suggestions for Authors

This systematic review aimed to comprehensively determine the sociodemographic, economic, and clinical factors contributing to the distribution of breast reconstruction for cancer patients, in order to improve health equity among the breast cancer population.

comments: 

1) line 173 and line 27- can omit the 'new' for the studies. can just mention that 70 studies were produced. 

2) one limitation of this study is that the results will only be applicable to the United states based on the search criteria. pls mention this limitation in the limitation section

3) line 261- leave a space for overtime

Comments on the Quality of English Language

minor edits needed

Author Response

  1. “New” removed from both lines 27 and 179
  2. Limitations were adjusted to “Literature was restricted to the English language and those published within the United States, potentially excluding additional work leading to language bias and findings applicable only to the United States.”
  3. “Overtime” was adjusted to “over time”.

Round 2

Reviewer 1 Report

Comments and Suggestions for Authors

I find the manuscript difficult to follow and not clear in multiple aspects, sometimes contradictory.

It is hard to follow not just for myself but likely for other readers of the journal as well.

Despite offering suggestions for improvement, some were not incorporated.

Based on these concerns, my recommendation is to reject the manuscript.